# Properties of Doubly Heavy Baryons

Zalak Shah *[ID], Amee Kakadiya, Keval Gandhi [ID] and Ajay Kumar Rai [ID]

Department of Physics, S.V. National Institute of Technology, Surat 395007, India; ameekakadiya@gmail.com (A.K.); kevalgandhi25194@gmail.com (K.G.); raiajayk@gmail.com (A.K.R.)
* Correspondence: zalak.physics@gmail.com

**Abstract:** We revisited the mass spectra of the $\Xi_{cc}^{++}$ baryon with positive and negative parity states using Hypercentral Constituent Quark Model Scheme with Coloumb plus screened potential. The ground state of the baryon has been determined by the LHCb experiment, and the anticipated excited state masses of the baryon have been compared with several theoretical methodologies. The transition magnetic moments of all heavy baryons $\Xi_{cc}^{++}$, $\Xi_{cc}^{+}$, $\Omega_{cc}^{+}$, $\Xi_{bb}^{0}$, $\Xi_{bb}^{-}$, $\Omega_{bb}^{-}$, $\Xi_{bc}^{+}$, $\Xi_{bc}^{0}$, $\Omega_{bc}^{0}$ are also calculated and their values are $-1.013$ $\mu_N$, $1.048$ $\mu_N$, $0.961$ $\mu_N$, $-1.69$ $\mu_N$, $0.73$ $\mu_N$, $0.48$ $\mu_N$, $-1.39$ $\mu_N$, $0.94$ $\mu_N$ and $0.710$ $\mu_N$, respectively.

**Keywords:** mass spectra; potential model; radiative transitions





## 1. Introduction

Hadron spectroscopy is a technique for understanding the dynamics of quark interactions in composite systems. The probable decays of the resonance states could be used to identify short-lived hadrons and missing excited states. Baryons are strongly interacting fermions made up of three quarks with an integer spin of $\frac{1}{2}$. The existence of degenerate levels of different charges with all the properties of isospin multiplets, quartets, triplets, doublets, and singlets is one of the most remarkable aspects of the baryon spectrum. The Eightfold geometrical design for mesons and baryons was introduced by Murray Gell-Mann in 1962 [1]. The baryon octet is made up of the eight lightest baryons that fit into a hexagonal arrangement with two particles in the center. The baryon decuplet is a triangular array of 10 particles. In addition, the antibaryon octet and decuplet exist, each with a different charge and strangeness. SU(4) group includes all of the baryons containing zero, one, two, or three heavy Q (charm or beauty) quarks with light (u, d and s) quarks. The representation shows totally symmetric 20-plet, the mixed symmetric 20′-plet and the total anti-symmetric $\bar{4}$ multiplet. The ground levels of SU(4) group multiplets are SU(3) decuplet, octet, and singlet, respectively. These baryon states can be further decomposed according to the heavy quark content inside. SU(3) is the group symmetry transformations of the 3-vector wavefunction that maintain the physical constraint that the total probability for finding the particle in one of the three possible states equals 1 [2]. The doubly heavy baryons are located on the second level of SU(4) multiplets with a combination of two heavy quarks (QQ) with one light quark (q). The two heavy quark combinations cc, bb and bc unifies with s quark in case of three doubly heavy $\Omega$ baryons, while for six doubly heavy $\Xi$ baryons light quarks u or d are combined.

In the last few years, many excited states of singly heavy baryons such as $\Lambda_c$, $\Xi_c$, $\Omega_c$, $\Lambda_b$, $\Sigma_b$, $\Xi_b$, $\Omega_b$ have been discovered with associated decay channels by worldwide experiments, namely, LHCb, BELLE, BARBAR, CDF, and CLEO. The experimentally discovered states are listed in Particle Data Group summary tables [3]. Many experiments have attempted to identify the doubly heavy baryons [4–7], but only a few ground states have been discovered so far. The specifics are listed below

SELEX experiment (in 2002): A ground state at 3520 MeV containing two charm quarks and a down quark in its decay mode $\Xi_{cc}^{+} \rightarrow pD^{+}K^{-}$ [8,9].

LHCb experiment (in 2018): $\Xi_{cc}^{++}$ with the mass $(3621.40 \pm 0.72 \pm 0.27 \pm 0.14)$ MeV and quark combination *ccu*. The decay mode of the experimental investigation is $\Xi_{cc}^{++} \rightarrow \Lambda_c^+ K^- \pi^+ \pi^-$ [10].

LHCb experiment (in 2021): $\Omega_{bc}^0$ and $\Xi_{bc}^0$ in the mass range from 6700 to 7300 MeV/$c^2$ are presented, using pp collision. These baryons are reconstructed in $\Lambda_c^+ \pi^-$ and $\Xi_c^+ \pi^-$ decay modes. No evidence of signal is found [11].

We can also observe the $\approx$100 MeV mass difference of the ground states of doubly charm $\Xi$ baryons in two experimental results (Selex and LHCb). Thus, in context with the recent experimental observation of $\Xi_{cc}^{++}$ baryon, we revisited the masses of radial and orbital excited states in present study using Hypercentral Constituent Quark Model. The model is used with Coulomb plus linear potential potential for the study of doubly heavy baryons [12,13]. To obtain the mass spectra, the Coulomb plus screened potential is employed with the first order correction, which gives a relativistic effect of order O(1/m) in this paper.

The paper is organized as follows: Following the introduction, Section 2 explains our Hypercentral constituent quark model with coulomb plus screened potential. The radial and orbital excited state masses and regge trajectory is discussed. Section 3 includes the magnetic moments and radiative transition of doubly heavy baryons, followed by a conclusion in Section 4.

## 2. Methodology

Hadron spectroscopy attempts to explore the excited mass spectrum along with the multiplet structure as well as spin-parity assignments. To understand the quark dynamics, it is essential to understand the quark confinement and asymptotic freedom first and then one can use any phenomenological approach to determine the properties of system. The Constituent Quark Model is based on a system of three quarks interacting by some potential and determine the different properties. It undertakes the baryon as a confined system of three quarks wherein the potential is the hypercentral one. The dynamics of three body system are taken care of using the Jacobi coordinates introduced as $\rho$ and $\lambda$ reducing it to two body parameters (see Figure 1). The Hypercentral constituent quark model(hCQM) scheme amounts to an average two-body potential for the three quark system over the hyper angle.

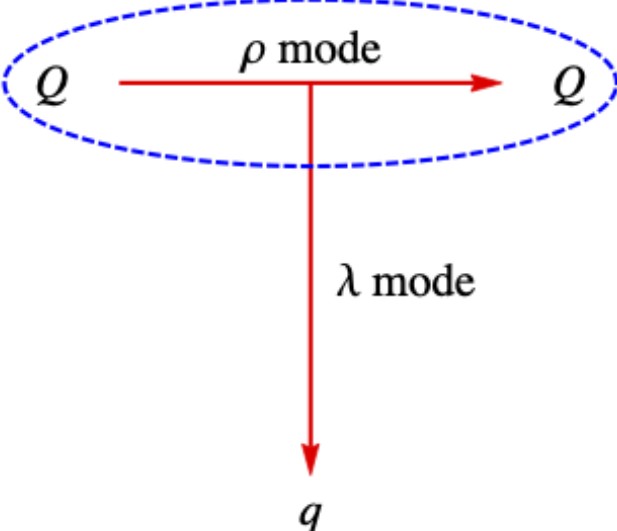

**Figure 1.** Doubly heavy baryon system with Jacobi coordinates $\rho$ and $\lambda$ mode.

The relevant degrees of freedom for the relative motion of the three constituent quarks provided by the relative Jacobi coordinates ($\vec{\rho}$ and $\vec{\lambda}$) are given by [14,15]

$$\vec{\rho} = \frac{1}{\sqrt{2}}(\vec{r_1} - \vec{r_2}) \qquad \vec{\lambda} = \frac{m_1\vec{r_1} + m_2\vec{r_2} - (m_1 + m_2)\vec{r_3}}{\sqrt{m_1^2 + m_2^2 + (m_1 + m_2)^2}} \tag{1}$$

The hyper radius $x$ is a collective variable gives a measure of dimension of the three-quark system and and the hyper angle $\xi$ reflects its deformation. We define hyper radius $x$ and hyper angle $\xi$ in terms of the absolute values $\rho$ and $\lambda$ of the Jacobi coordinate [16]

$$x = \sqrt{\rho^2 + \lambda^2}, \qquad \xi = arctan\left(\frac{\rho}{\lambda}\right) \tag{2}$$

The Hamiltonian of three body baryonic system in the hCQM is then expressed as [17]

$$H = \frac{P_x^2}{2m} + V(x) \tag{3}$$

The potential would be the Coulomb plus screened potential with the first order correction. The form of the potential is [18,19],

$$V(x) = V_{SI}(x) + \left(\frac{1}{m_\rho} + \frac{1}{m_\lambda}\right)V^{(1)}(x) + V_{SD}(x) \tag{4}$$

where the effective spin-independent static potential $V_{SI}(x)$ is simply the sum of Lorentz vector $V_V(x)$ and Lorentz scalar $V_S(x)$ terms.

$$V_{SI}(x) = V_{Col}(x) + V_{conf}(x) \tag{5}$$

$$V_{Col}(x) = -\frac{2}{3}\frac{\alpha_s}{x}, V_{conf}(x) = a\left(\frac{1 - e^{-\mu x}}{\mu}\right), V^{(1)}(x) = C_F C_A \frac{\alpha_s^2}{4x^2}, \tag{6}$$

$$V_{SD}(x) = V_{SS}(x)(S_\rho \cdot S_\lambda) + V_{\gamma S}(x)(\gamma \cdot S) + V_T(x)[S^2 - \frac{3(S \cdot x)(S \cdot x)}{x^2}] \tag{7}$$

where $V_{Col}(x)$ is the QCD potential interacting between two quarks of three quark baryonic system which is non-relativistic. $\alpha_s$ is a strong coupling constant. $a$ is the string tension and the constant $\mu$ is the screening factor. For $x \ll \frac{1}{\mu}$, the screened potential is behaving like a linear potential $ax$ and for $x \gg \frac{1}{\mu}$ it becomes a constant $\frac{a}{\mu}$. Therefore, it is interesting to study the mass spectroscopy with screened potential which gives the masses of the higher excited states lesser compared to the linear potential [20,21]. $V^1(x)$ is the first order correction in terms of Casimir charges of the fundamental and adjoint representation, coupling constant and hyper radius. The $V_{SD}(x)$ accounts for the spin-dependent terms leading to hyperfine interactions such as the spin–spin term $V_{SS}(x)$, the spin-orbit term $V_{\gamma S}(x)$ and tensor term $V_T(x)$ [22,23]. We have numerically solved the six-dimensional *Schrödinger* equation using Mathematica notebook.

$$\left[\frac{-1}{2m}\frac{d^2}{dx^2} + \frac{\frac{15}{4} + \gamma(\gamma + 4)}{2mx^2} + V(x)\right]\phi_\gamma(x) = E\phi_\gamma(x) \tag{8}$$

The spin average masses are determined by taking a summation of model quark masses with its binding energy,

$$M_{SA} = E + m_{q1} + m_{q2} + m_{q3}. \tag{9}$$

Thus, the total mass can be determined for doubly heavy baryon system would be

$$M_{total} = M_{SA} + V(x) - V_{SI}(x). \tag{10}$$

The mass of doubly heavy baryons are studied within the various models like hypercentral method [24], relativistic model [25,26], varational model [27], QCD sum rules [28–31], lattice QCD [32], Hamiltonian Model [33], Regge phenomenology [34,35] and Chiral quark model [36], the three-body Faddeev method [37], heavy quark symmetry [38], diquak approximation [39], Lattice QCD [40], and many more. The excited states predictions are mentioned in Table 1. We calculated the masses of ground state (1S), radial excited states (2S–5S) and orbital excited states (1P–5P, 1D–4D, 1F–2F) with all isospin splittings. The results are shown in Tables 2–4.

**Table 1.** Excited state masses by various theoretical approaches (in GeV).

| $J^P$ | [24] | [27] | [33] | [25] | [37] | [38] | [30] | [34] |
|---|---|---|---|---|---|---|---|---|
| $\frac{1}{2}^+$ | 3.901 | 4.029 | 4.079 | 3.910 | 3.976 | 4.030 | 4.03 | |
| $\frac{1}{2}^+$ | 4.118 | | 4.206 | 4.154 | | | | |
| $\frac{3}{2}^+$ | 3.958 | 4.042 | 4.114 | 4.027 | 4.025 | 4.078 | 3.962 | |
| $\frac{3}{2}^+$ | 4.211 | | 4.131 | | | | | |
| $\frac{1}{2}^-$ | 3.847 | 3.910 | 3.947 | 3.838 | 3.880 | 4.073 | 4.03 | |
| $\frac{3}{2}^-$ | 3.830 | 3.921 | 3.949 | 3.959 | | 4.079 | | 3.786 |
| $\frac{5}{2}^+$ | 4.019 | 4.027 | 4.187 | | | 4.393 | | 4.089 |
| $\frac{7}{2}^-$ | 4.150 | | | | | | | 4.267 |

**Table 2.** Radial excited state masses (in GeV).

| $J^P$ | State | Our | $J^P$ | State | Our |
|---|---|---|---|---|---|
| $\frac{1}{2}^+$ | 2S | 4.046 | $\frac{3}{2}^+$ | 2S | 4.082 |
| $\frac{1}{2}^+$ | 3S | 4.448 | $\frac{3}{2}^+$ | 3S | 4.466 |
| $\frac{1}{2}^+$ | 4S | 4.795 | $\frac{3}{2}^+$ | 4S | 4.805 |
| $\frac{1}{2}^+$ | 5S | 5.103 | $\frac{3}{2}^+$ | 5S | 5.109 |

**Table 3.** Orbital excited state masses (in GeV).

| State | S | $J^P$ | Our |
|---|---|---|---|
| 1P | $\frac{1}{2}$ | $\frac{1}{2}^-$ | 3.978 |
| | $\frac{1}{2}$ | $\frac{3}{2}^-$ | 4.026 |
| | $\frac{3}{2}$ | $\frac{1}{2}^-$ | 3.982 |
| | $\frac{3}{2}$ | $\frac{3}{2}^-$ | 3.973 |
| | $\frac{3}{2}$ | $\frac{5}{2}^-$ | 3.961 |
| 2P | $\frac{1}{2}$ | $\frac{1}{2}^-$ | 4.373 |
| | $\frac{1}{2}$ | $\frac{3}{2}^-$ | 4.366 |
| | $\frac{3}{2}$ | $\frac{1}{2}^-$ | 4.125 |
| | $\frac{3}{2}$ | $\frac{3}{2}^-$ | 4.109 |
| | $\frac{3}{2}$ | $\frac{5}{2}^-$ | 4.361 |

**Table 3.** *Cont.*

| State | S | $J^P$ | Our |
|---|---|---|---|
| 3P | $\frac{1}{2}$ | $\frac{1}{2}^-$ | 4.721 |
| | $\frac{1}{2}$ | $\frac{3}{2}^-$ | 4.716 |
| | $\frac{3}{2}$ | $\frac{1}{2}^-$ | 4.725 |
| | $\frac{3}{2}$ | $\frac{3}{2}^-$ | 4.719 |
| | $\frac{3}{2}$ | $\frac{5}{2}^-$ | 4.711 |
| 4P | $\frac{1}{2}$ | $\frac{1}{2}^-$ | 5.033 |
| | $\frac{1}{2}$ | $\frac{3}{2}^-$ | 5.029 |
| | $\frac{3}{2}$ | $\frac{1}{2}^-$ | 5.035 |
| | $\frac{3}{2}$ | $\frac{3}{2}^-$ | 5.031 |
| | $\frac{3}{2}$ | $\frac{5}{2}^-$ | 5.025 |
| 5P | $\frac{1}{2}$ | $\frac{1}{2}^-$ | 5.316 |
| | $\frac{1}{2}$ | $\frac{3}{2}^-$ | 5.312 |
| | $\frac{3}{2}$ | $\frac{1}{2}^-$ | 5.317 |
| | $\frac{3}{2}$ | $\frac{3}{2}^-$ | 5.314 |
| | $\frac{3}{2}$ | $\frac{5}{2}^-$ | 5.309 |

**Table 4.** Orbital excited state masses (in GeV).

| State | S | $J^P$ | Our |
|---|---|---|---|
| 1D | $\frac{1}{2}$ | $\frac{3}{2}^+$ | 4.163 |
| | $\frac{1}{2}$ | $\frac{5}{2}^+$ | 4.100 |
| | $\frac{3}{2}$ | $\frac{1}{2}^+$ | 4.118 |
| | $\frac{3}{2}$ | $\frac{3}{2}^+$ | 4.112 |
| | $\frac{3}{2}$ | $\frac{5}{2}^+$ | 4.103 |
| | $\frac{3}{2}$ | $\frac{7}{2}^+$ | 4.092 |
| 2D | $\frac{1}{2}$ | $\frac{3}{2}^+$ | 4.420 |
| | $\frac{1}{2}$ | $\frac{5}{2}^+$ | 4.412 |
| | $\frac{3}{2}$ | $\frac{1}{2}^+$ | 4.427 |
| | $\frac{3}{2}$ | $\frac{3}{2}^+$ | 4.422 |
| | $\frac{3}{2}$ | $\frac{5}{2}^+$ | 4.415 |
| | $\frac{3}{2}$ | $\frac{7}{2}^+$ | 4.407 |
| | $\frac{1}{2}$ | $\frac{3}{2}^+$ | 4.696 |
| 3D | $\frac{1}{2}$ | $\frac{5}{2}^+$ | 4.691 |
| | $\frac{3}{2}$ | $\frac{1}{2}^+$ | 4.701 |
| | $\frac{3}{2}$ | $\frac{3}{2}^+$ | 4.698 |
| | $\frac{3}{2}$ | $\frac{5}{2}^+$ | 4.693 |
| | $\frac{3}{2}$ | $\frac{7}{2}^+$ | 4.687 |
| 4D | $\frac{1}{2}$ | $\frac{3}{2}^+$ | 4.945 |
| | $\frac{1}{2}$ | $\frac{5}{2}^+$ | 4.942 |
| | $\frac{3}{2}$ | $\frac{1}{2}^+$ | 4.950 |
| | $\frac{3}{2}$ | $\frac{3}{2}^+$ | 4.947 |
| | $\frac{3}{2}$ | $\frac{5}{2}^+$ | 4.944 |
| | $\frac{3}{2}$ | $\frac{7}{2}^+$ | 4.939 |

**Table 4.** *Cont.*

| State | S | $J^P$ | Our |
|---|---|---|---|
| 1F | $\frac{1}{2}$ | $\frac{5}{2}^-$ | 4.235 |
|  | $\frac{1}{2}$ | $\frac{7}{2}^-$ | 4.227 |
|  | $\frac{3}{2}$ | $\frac{3}{2}^-$ | 4.245 |
|  | $\frac{3}{2}$ | $\frac{5}{2}^-$ | 4.238 |
|  | $\frac{1}{2}$ | $\frac{7}{2}^-$ | 4.230 |
|  | $\frac{1}{2}$ | $\frac{9}{2}^-$ | 4.219 |
| 2F | $\frac{1}{2}$ | $\frac{5}{2}^-$ | 4.497 |
|  | $\frac{1}{2}$ | $\frac{7}{2}^-$ | 4.491 |
|  | $\frac{3}{2}$ | $\frac{3}{2}^-$ | 4.504 |
|  | $\frac{3}{2}$ | $\frac{5}{2}^-$ | 4.499 |
|  | $\frac{1}{2}$ | $\frac{7}{2}^-$ | 4.493 |
|  | $\frac{1}{2}$ | $\frac{9}{2}^-$ | 4.485 |

The mass spectra (from S state to F state) of $\Xi_{cc}^{++}$ baryon is determined by the works in [12,24] using hCQM. In present study, we study the excited states using Coulomb plus screen confinement, so that the masses are higher than the previous study. We can observe that, the masses of our first radial excited state (2S) with $J^P$ values, $\frac{1}{2}^+$ and $\frac{3}{2}^+$ are in accordance with the works in [27,30,33,38]. The first orbital excited state(1P) masses are near with the the works in [27,33] among all, while the 1D state with $J^P$ value, $\frac{5}{2}^+$ shows 84, 76, 84, and 14 MeV diffrence with the aforementioned works shown in Table 1. Our 1F state with $J^P$ value, $\frac{7}{2}^-$ show only 4 MeV difference with our obtained mass. Comparing with the rest of the excited states of Table 1, the values have far differences.

The Regge trajectory is useful to determine the unknown quantum numbers from the slopes ($\alpha$) and intercepts ($\alpha_0$) [12,13,41–44]:

$$J = \alpha M^2 + \alpha_0 \tag{11}$$

The plot of total angular momentum J against the square of resonance mass $M^2$ are drawn based on calculated data (of Tables 2 and 3). The Regge trajectory is drawn for 1S–3S, 1P–3P, 1D–3D, and 1F–2F states with linear fit in Figure 2. As we reach to the higher excited states, the screening effective can be easily identified.

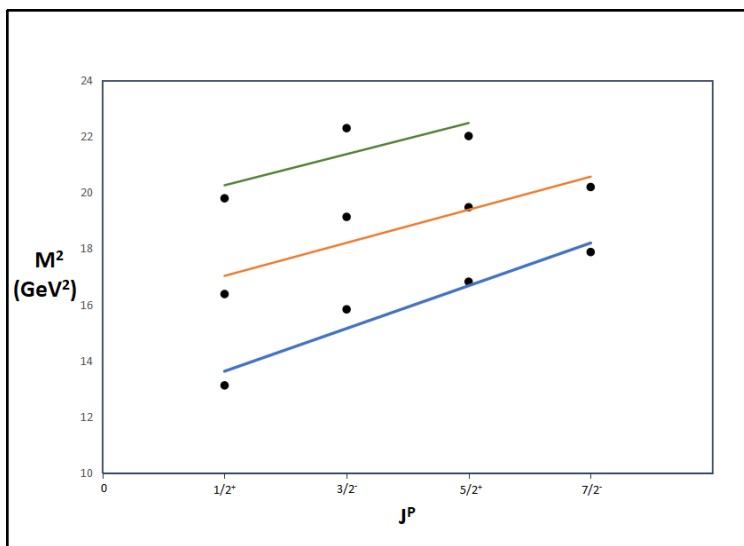

**Figure 2.** The regge trajectory of $\Xi_{cc}^{++}$ baryon in (J, $M^2$) plane.

### 3. Magnetic Moments and Radiative Transitions

There are different approaches in investigation of the structure of hadrons, especially, the most promising one in this direction is the study of the electromagnetic properties of the hadrons and their excitations. The electromagnetic transitions of doubly heavy baryons have been evaluated in the heavy baryon chiral perturbation theory [45,46], light-cone sum rules [47], the relativistic quark model [48], a nonrelativistic approach [49–54], bag Model [55], and many more.

The magnetic moment of the baryons with bound quarks are given in the form of a charge ($e_i$) and the spin of the respective constituent quark corresponds to the spin flavor wave function ($\sigma_i$) based on the non-relativistic hypercentral constituent quark model are [49,50]

$$\mu_B = \sum_i \langle \phi_{sf} | \mu_{iz} | \phi_{sf} \rangle \tag{12}$$

where $\mu_i = \frac{e_i \sigma_i}{2m_i^{eff}}$ and $i$ = u, d, s, c, b. To obtain magnetic moments of the baryons, we need to calculate their effective masses first. As the combination of quarks in baryon changes, its binding interaction affects and $m_i^{eff}$ differs. The effective mass for each of the constituting quark $m_i^{eff}$ can be defined as

$$m_i^{eff} = m_i \left( 1 + \frac{\langle H \rangle}{\sum_i m_i} \right) \tag{13}$$

where $\langle H \rangle$ = E + $\langle V_{spin} \rangle$. We adopt the $m_u$ = 338 MeV, $m_d$ = 350 MeV, $m_s$ = 500 MeV, $m_c$ = 1295 MeV, $m_b$ = 4670 MeV as a constituent quark masses. The expressions for magnetic moments of doubly heavy baryons are

- For doubly charmed baryons, $J^P = \frac{1}{2}^+$

$$\frac{4}{3}\mu_c - \frac{1}{3}\mu_q \tag{14}$$

- For doubly bottom baryons, $J^P = \frac{1}{2}^+$

$$\frac{4}{3}\mu_b - \frac{1}{3}\mu_q \tag{15}$$

- For doubly charm-bottom baryons, $J^P = \frac{1}{2}^+$

$$\frac{2}{3}\mu_b + \frac{2}{3}\mu_c - \frac{1}{3}\mu_q \tag{16}$$

- For doubly charmed baryons, $J^P = \frac{3}{2}^+$

$$2\mu_c + \mu_q \tag{17}$$

- For doubly bottom baryons, $J^P = \frac{3}{2}^+$

$$2\mu_b + \mu_q \tag{18}$$

- For doubly charm-bottom baryons, $J^P = \frac{3}{2}^+$

$$\mu_b + \mu_c + \mu_q \tag{19}$$

Here, the $q$ is light quark of the system. The magnetic moments of double heavy baryons are computed using Equations (4)–(11) in terms of the nuclear magnetons and spin 1/2 and spin 3/2. The numerical results are shown in Table 5 for all doubly heavy baryon system. The obtained results are in accordance with the works in [53–62].

**Table 5.** The magnetic moments of doubly heavy baryons (in terms of nuclear magnaton μ$_N$).

| Baryon | $J^P$ | Magnetic Moment | Baryon | $J^P$ | Magnetic Moment |
|--------|-------|-----------------|--------|-------|-----------------|
| $\Xi_{cc}^{+}$ | $\frac{1}{2}^{+}$ | 0.784 | $\Xi_{cc}^{+}$ | $\frac{3}{2}^{+}$ | 0.068 |
| $\Xi_{cc}^{++}$ | $\frac{1}{2}^{+}$ | 0.031 | $\Xi_{cc}^{++}$ | $\frac{3}{2}^{+}$ | 2.218 |
| $\Xi_{bb}^{-}$ | $\frac{1}{2}^{+}$ | 0.196 | $\Xi_{bb}^{-}$ | $\frac{3}{2}^{+}$ | −1.737 |
| $\Xi_{bb}^{0}$ | $\frac{1}{2}^{+}$ | −0.663 | $\Xi_{bb}^{0}$ | $\frac{3}{2}^{+}$ | −1.607 |
| $\Omega_{cc}^{+}$ | $\frac{1}{2}^{+}$ | 0.692 | $\Omega_{cc}^{+}$ | $\frac{3}{2}^{+}$ | 0.285 |
| $\Omega_{bb}^{-}$ | $\frac{1}{2}^{+}$ | 0.108 | $\Omega_{bb}^{-}$ | $\frac{3}{2}^{+}$ | −1.239 |
| $\Xi_{bc}^{0}$ | $\frac{1}{2}^{+}$ | 0.527 | $\Xi_{bc}^{0}$ | $\frac{3}{2}^{+}$ | −0.448 |
| $\Xi_{bc}^{+}$ | $\frac{1}{2}^{+}$ | −0.304 | $\Xi_{bc}^{+}$ | $\frac{3}{2}^{+}$ | 2.107 |

The radiative transition moment can be expressed as

$$\mu_{B^* \to B\gamma} = \langle B^* | \mu_{B^* \to B\gamma} | B \rangle \tag{20}$$

The transition magnetic moment ($\mu_i$) of the constituent quarks are computed using the spin flavour wave functions of the initial and final baryons are [49,50]

$$\mu_i = \langle \phi_{sf} | \frac{e_i}{2m_i^{eff}} \sigma_{iz} | \phi_{sf} \rangle \tag{21}$$

Here, $m_i^{eff}$ is the effective mass of the constituent quarks within the baryons. In order to evaluate the $B_{\frac{3}{2}^+} \to B_{\frac{1}{2}^+}\gamma$ transition magnetic moments, we take the geometric mean of effective quark masses of the constituent quarks of initial and final states baryons. The spin-3/2 to spin-1/2 doubly heavy baryon transition magnetic moments in the quark model can be determined as follows [45].

- For doubly charmed baryons,

$$\frac{4}{3\sqrt{2}}(\mu_Q - \mu_q) \tag{22}$$

- For doubly bottom baryons,

$$\frac{2\sqrt{2}}{3}(\mu_Q - \mu_q) \tag{23}$$

- For doubly charm-bottom baryons,

$$\frac{\sqrt{2}}{3}(\mu_Q + \mu_Q - 2\mu_q) \tag{24}$$

Here, $Q$ and $q$ are heavy and light quarks, respectively. Using the above relations, the transition magnetic moments are calculated for all doubly heavy baryons, tabulated in Table 6. These baryons are transisted from spin-3/2 to spin-1/2 state. Their masses are obtained from the works in [12,13]. Using these transition magnetic moments, the radiative decays can also be identified. We can see that, the results are in accordance with the other theoretical prediction in Table 6. Recently, the work in [46] has also investigated the values of magnetic moments and radiative transitions of doubly charm baryons with spin-1/2.

**Table 6.** Transision Magnetic Moments of doubly charm, doubly bottom and doubly charm-bottom baryons (in units of $\mu_N$).

| Process | Our Results | [45] | [63] | [64] | [57] | [55] |
|---|---|---|---|---|---|---|
| $\Xi_{cc}^{++*} \rightarrow \Xi_{cc}^{++}$ | $-1.01$ | $-1.40$ | 0.17 | $-0.47$ | 1.35 | $-0.787$ |
| $\Xi_{cc}^{+*} \rightarrow \Xi_{cc}^{+}$ | 1.048 | 1.23 | 0.86 | 0.98 | 1.06 | 0.945 |
| $\Omega_{cc}^{+*} \rightarrow \Omega_{cc}^{+}$ | 0.96 | 0.90 | 0.84 | 0.59 | 0.88 | 0.789 |
| $\Xi_{bb}^{0*} \rightarrow \Xi_{bb}^{0}$ | $-1.69$ | $-1.81$ | | | | |
| $\Xi_{bb}^{-*} \rightarrow \Xi_{bb}^{-}$ | 0.73 | 0.81 | | | | |
| $\Omega_{bb}^{-*} \rightarrow \Omega_{bb}^{-}$ | 0.48 | 0.48 | | | | |
| $\Xi_{bc}^{+*} \rightarrow \Xi_{bc}^{+}$ | $-1.39$ | $-1.61$ | | | | |
| $\Xi_{bc}^{0*} \rightarrow \Xi_{bc}^{0}$ | 0.94 | 1.02 | | | | |
| $\Omega_{bc}^{0*} \rightarrow \Omega_{bc}^{0}$ | 0.71 | 0.69 | | | | |

## 4. Conclusions

The baryon containing two heavy quarks (*cc*) with a light (*u*) quark is reviewed. The ground, radial, and orbital excited masses are calculated using the Hypercentral Constituent Quark Model (hCQM). The Coulomb plus screened potential is employed with the first order correction. The obtained results are compared and we conclude that the excited state masses of 2S, 1P, 1D, and 1F states are in order with the other models predictions. The magnetic moments are also identified for spin 1/2 and spin 3/2 states of all doubly heavy baryons; $\Xi_{cc}^{++}$, $\Xi_{cc}^{+}$, $\Xi_{bb}^{-}$, $\Xi_{bb}^{0}$, $\Xi_{bc}^{+}$, $\Xi_{bc}^{0}$, $\Omega_{cc}^{+}$, $\Omega_{bb}^{-}$, $\Omega_{bc}^{0}$. Furthermore, the transition magnetic moments are determined of the ground states. The spin-3/2 to spin-1/2 doubly heavy baryon transitions have yet to be studied experimentally. Future experiments and other theoretical models will get benefit from this research in identifying baryonic states from resonances and transitions. We would like to calculate the radiative decays and semi leptonic decays for double heavy baryons in future work.

**Author Contributions:** Conceptualization and writing Z.S., methodology K.G., formal analysis A.K., Supervision A.K.R. All authors have read and agreed to the published version of the manuscript.

**Funding:** The APC was funded by the organizers of XXXII International (ONLINE) Workshop on High Energy Physics Hot problems of Strong Interactions.

**Institutional Review Board Statement:** Not applicable.

**Informed Consent Statement:** Not applicable.

**Acknowledgments:** Z. Shah thanks the organizers of the XXXII International Workshop on High Energy Physics "Hot problems of Strong Interactions", especially V. Petrov, R. Zhokhov and A. Luchinsky, for the invitation to participate in such an interesting and productive meeting.

**Conflicts of Interest:** The authors declare no conflict of interest.

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
