# Peer review of "Properties of Doubly Heavy Baryons"

_universe, doi:10.3390/universe7090337_

Round 1

Reviewer 1 Report

Zalak Shah et. al. might define some quantities such as "hyper angle' on page 2

Author Response

The hyper radius $x$ is a collective coordinate and therefore the hypercentral potential contains also the three-body effects. and the hyper angle $\xi$ reflects its deformation. We define hyper radius x and hyper angle $\xi$ in terms of the absolute values $\rho$ and $\lambda$ of the Jacobi coordinate \cite{Fabre1988} \begin{equation} x= \sqrt{\rho^2 + \lambda^2}, ~~~~~~ \xi= arctan \left( \frac{\rho}{\lambda}\right) \end{equation}
It is define on page-3 of revised manuscript.

Reviewer 2 Report

Referee report on the manuscript # universe-1353022

by Zalak Urjit Shah, Amee Kakadiya, Keval Kumar Gandhi

and Ajay Kumar Rai

entitled "Properties of Doubly Heavy Baryons".

In this manuscript the authors present the study of mass spectra of

the baryons containing two heavy quarks and one light up quark. The

baryon mass spectra are calculated within the Hypercentral Constituent

Quark Model (hCQM) with the Coulomb plus screened potential. Additional

first order correction is also considered. The calculations have been

performed in Matematica package. The results are listed in six Tables

and compared to predictions of other models.

It is shown that the masses of excited states of 2S, 1P, 1D and 1F are

in fair agreement with the other model predictions. Magnetic moments

are also identified for all doubly heavy baryons with spins 1/2 and

3/2. The transition magnetic moments are also determined for the

ground states. These results might be helpful for experimental study

of doubly heavy baryon transitions from the spin-3/2 to spin-1/2

states.

There are two issues to fix before the formal acceptance of the

manuscript for publication.

1. The presentation of the method has to be more detailed.

Note, that the description of many parameters entering equations

(3)-(6) is missing (e.g., \tau, \mu, S  etc).

The employed algorithm of calculation of baryon masses is also

missing. Otherwise, the reader may be confused where the numbers

indicated in the Tables are taken from.

2. There are many misprints in the text, e.g., "Coloumb" (line 2)

and "Colomb" (l.155) instead of "Coulomb", "differnt" (line 67),

"where, The ..." (l.78), "stateto" (l.96), and so forth. Also, the

sentence in lines 75-76 should be corrected.

Author Response

Thank you very much for your valuable suggestions.

(1) As per your suggestions, section -2 (page-3) is modified. The parameters are described; the equations and references are added. The added Equation (9-10) are used for the mass spectra.

(2) We are sorry for the grammatical misprints. We correct it.

(3) Eq. 11 and some references are added in the revised manuscript.

This manuscript is a resubmission of an earlier submission. The following is a list of the peer review reports and author responses from that submission.

Round 1

Reviewer 1 Report

A sound justification about the novelties of this work regarding Refs. [12] and [13] should be provided. If it were not so this paper cannot be published.

Author Response

The properties of doubly heavy charm, bottom, and charm-bottom baryons are discussed in this study. We need to talk about baryon masses first, which is why we only discuss about ground states and promising excited states like 2S, 1P, 1D, and 1F. We can see that experiments like LHCb and BELLE provide the masses and decays of these states, so we're attempting to figure out what they're like. We have calculated the magnetic moments and transition magnetic moments of these baryons using these masses. The acquired results are compared to other theoretical predictions, which may aid future studies in locating resonances. We have previously identified entire spectra from 1S-5S, 1P-5P, 1D-4D, and 1F-2F states with all isospin splittings and plotted Regge trajectories in the (n, M2) and (J, M2) planes using these masses. We also revisited the doubly heavy charm baryon spectra (Proceedings of the DAE Symp. on Nucl. Phys. 64 (2019)) and we incorporate those results in the present study.

Reviewer 2 Report

The authors calculate the properties - namely masses and magnetic
moments - of doubly heavy baryons, with either two charm, two beauty
or one charm and one beauty quark.
They utilize the hypercentral consituent quark model (hCQM) to study
the ground states and some excited states of the $\Sigma_{cc}^+$,
$\Sigma_{bb}^-$, $\Xi_{cc}^{++}$and $\Xi_{bb}^{--}$ baryons, adding
to existing literature in which the same or different approaches are
utilized.

The results the authors present are in line with the results of the
other works they compare to, both for the masses determination, as well
as for the magnetic moments.

Although already present in previous works that the authors cite
(e.g. Refs. [12,13]), I believe a more thorough description of the model
they implement would benefit the paper.

In any case, the major concerns I have about this work concern the masses
determinations, and are the following:
1) A previous work (Adv.High Energy Phys. 2018 (2018) 1326438, arXiv 1807.06800) seems to address the determination of double heavy baryons' masses with an extremely similar approach. The work is not cited in this manuscript, hence I do not know whether the authors are aware of it. Can the authors comment on this?
2) More importantly, all the results presented in this work for the baryon
masses, summarized in Tables 2-5, are already present in Refs.[12,13]. In that
case, they are presented as corresponding to a scenario where no first-order
correction to the potential is considered, while in this case the authors state such correction is included. I believe it is crucial that the authors clarify this issue, in particular the presence of previously published results.

I would consider to suggest publication for this work upon examining the
response the authors provide, especially to the second point.

Author Response

We are very thankful to your comments. We will definitely add the description of the model in revised MS. Thank you for suggesting the (Adv.High Energy Phys. 2018 (2018) 1326438, arXiv 1807.06800) article, we compare the results in Table 4-5. 

The properties of doubly heavy charm, bottom, and charm-bottom baryons are discussed in this study. We need to talk about baryon masses first, which is why we only discuss about ground states and promising excited states like 2S, 1P, 1D, and 1F. We can see that experiments like LHCb and BELLE provide the masses and decays of these states, so we're attempting to figure out what they're like. We have calculated the magnetic moments and transition magnetic moments of these baryons using these masses. The acquired results are compared to other theoretical predictions, which may aid future studies in locating resonances. We have previously identified entire spectra from 1S-5S, 1P-5P, 1D-4D, and 1F-2F states with all isospin splittings and plotted Regge trajectories in the (n, M2) and (J, M2) planes using these masses. We also revisited the doubly heavy charm baryon spectra (Proceedings of the DAE Symp. on Nucl. Phys. 64 (2019)) and we incorporate those results in the present study.

Reviewer 3 Report

line 56:Which two experimental results?

Eq(3) give V^0, V^(1), V_{SD}

Figure 2:Why 5 masses?

PDG(2020):\Omega_cc^+ \simeq 3.62 GeV (not 4.03)

Figure 3:Why 4 masses?

Section 3:What is "our model"

  Table 6:define \mu_N

Conclusion and 2.Methodology:define hCQM used in manuscript

Author Response

pls check attached file

Round 2

Reviewer 1 Report

Although the style and presentation issues have been partially corrected, my main concern regarding the paper being an original research is maintained. My opinion has not changed.

Reviewer 2 Report

I acknowledge the fact that the authors expanded the introduction to
provide a better description of the model utilized in this work.

However, the authors have failed to provide a justifications for why
the results summarized in this manuscript (collected in Tables 2-5), 
are already present in Refs.[12,13].

For this reason, I cannot recommend the present work for publication.

Reviewer 3 Report

A major problem is that in Tables units are not included

In Tables 2, 3, 4 and 5 there are no units for masses.

In Ref [1], Particle Data Group--Particle Physics Booklet

masses are in units of MeV.

In Tables 6 & 7 there are no units for Magnetic Moments.

The standard unit is J/T=Joule/Tesla